# Precision Surgery of Colorectal Liver Metastases in the Current Era: A Systematic Review

**DOI:** 10.3390/cancers15072083

**Published:** 2023-03-31

**Authors:** Dimitrios Papaconstantinou, Nikolaos Pararas, Anastasia Pikouli, Constantinos Nastos, Anestis Charalampopoulos, Dionysios Dellaportas, George Bagias, Emmanouil Pikoulis

**Affiliations:** Third Department of Surgery, Attikon University Hospital, 15772 Athens, Greece

**Keywords:** RAS, colorectal cancer, liver resection, metastases, R1

## Abstract

**Simple Summary:**

Colorectal cancer is among the most commonly encountered human malignancies, with up to a quarter of total patients exhibiting metastatic spread to the liver at some point during the disease course. For these patients, curative-intent liver resection yields the optimal oncological outcomes, ensuring long-term survival in the majority of cases. However, taking into account that liver resections are accompanied by considerable postoperative complication rates, treatment approaches should be individualized to avoid unnecessary exposure to morbid therapies. To that end, the study of tumor biology, especially in terms of RAS mutational status, is proving a very helpful adjunct in identifying patient subsets that derive the most benefit from this aggressive treatment modality. The concept of precision surgery in CRLM revolves around tailoring the aggressiveness of the employed treatments depending on tumor features, while avoiding surgery in patients not expected to derive any benefit from it.

**Abstract:**

Liver resection for colorectal liver metastases (CRLM) is widely considered the treatment with the highest curative potential. However, not all patients derive the same oncological benefit, underlining the need for better patient stratification and treatment allocation. In this context, we performed a systematic review of the literature to determine the role of RAS status in selecting the optimal surgical strategy. Evidence comparing anatomical with non-anatomical resections depending on RAS mutational status was scarce and conflicting, with two studies reporting superiority in mutated RAS (mutRAS) patients and two studies reporting equivalent outcomes. The rate of incomplete microscopic resection (R1) was found to be increased among mutRAS patients, possibly due to higher micrometastatic spread lateral to the primary lesion. The impact of R1 resection margins was evaluated separately for mutRAS and wild-type patients in three studies, of which, two indicated an additive detriment to long-term survival in the former group. In the current era of precision surgery, RAS status can be utilized to predict the efficacy of liver resection in the treatment of CRLM, avoiding a potentially morbid operation in patients with adverse tumor profiles.

## 1. Introduction

Among patients with colorectal cancer (CRC), metastatic spread to the liver (CRLM) is the leading cause of cancer-related mortality [1]. Liver resection has been considered as the treatment with the highest curative potential, with the reported 5-year survival ranging from 40 to 60% and 10-year survival ranging from 15 to 25% [2,3]. Over the past several decades, the criteria defining resectability for CRLM have expanded to include any patient in whom disease can be cleared, while leaving behind an adequate future liver remnant [4]. The oncological benefit of liver resection compared to chemotherapy or radiofrequency alone has been previously highlighted in several studies [5,6].

However, the high incidence of tumor recurrence after curative liver resection in CRLM remains an unresolved issue, with 5-year recurrence rates being as high as 70–80% [7]. A major determinant of disease reemergence is the microscopic completeness of the resection, as determined by histopathology [8]. Despite this, a debate exists regarding the appropriate margin width, with some investigators suggesting that margins wider than 1 mm may be associated with enhanced oncologic outcomes, possibly due to eradication of micrometastases or satellite lesions [9,10].

In the current era of precision medicine, the RAS gene has drawn attention for its role in CRC carcinogenesis. Mutations are encountered in up to 33–50% of all CRC patients [11], and carry a worse prognosis, with higher recurrence rates and suboptimal responses to chemotherapy [12,13]. In this regard, RAS mutated CRLM exhibit a more aggressive disease phenotype with some studies indicating an association between narrower resection margins and RAS mutational status [14]. Conceivably, RAS-mutated tumors are associated with wider infiltration patterns and exhibit a larger propensity for incomplete resection and worse local and overall recurrence rates [15]. In this context, the value of anatomic resections in patients with CRLM could be reexamined depending on tumor biology. More aggressive disease types may be optimally managed by more aggressive treatments and following this line of thought, anatomical resections may be expected to optimize recurrence outcomes, at least on the local level [16].

With RAS mutations being at the forefront of CRLM management, we sought to investigate the implications that their presence carries in patients with resectable tumors. In the present systematic review of the literature, we aim to evaluate the role of anatomic resections in reducing liver-specific disease recurrences and examine the impact of resection margin positivity depending on RAS mutational status.

## 2. Materials and Methods

### 2.1. Literature Search and Strategy

A systematic literature search of the Medline, Embase, Web of Science, CINAHL and CENTRAL databases was undertaken using the keywords “colorectal neoplasms”, “liver metastases”, “liver resection”, “hepatectomy”, “ras”, “kras”, “nras” synthesized into a search string using the Boolean operators AND/OR as appropriate for each database. After excluding duplicate studies, the abstract list generated by the search algorithm was independently screened by two authors (AP, GB) for relevant articles. All potentially eligible studies were reviewed in full text by two authors (AP, GB) while a third author (DP) acted as a referee in cases of disagreement. The reference lists of studies evaluated for inclusion were further manually screened using the snowballing technique to identify additional eligible studies.

The present systematic review was conducted according to PRISMA guidelines [17].

### 2.2. Outcomes of Interest, Data Extraction and Synthesis

The present systematic review was conducted as a three-part analysis; the first part focuses on outcomes following anatomical versus non-anatomical liver resections depending on patient RAS status. The second part relates to the incidence of microscopically incomplete resections (R1) depending on RAS status. The third part is focused on the impact of RAS status on survival, following a microscopically incomplete resection (R1).

Primary outcomes of interest were the overall survival (OS), the liver-specific disease-free survival (LS-DFS), and patient RAS status, as reported in eligible studies. Secondary outcomes of interest the tumor characteristics, patient demographics, and prognosticators for poor survival. All data extracted from eligible studies were inserted in standardized excel spreadsheets (Microsoft, Redmond, WA, USA) by two authors (CN, DD), with another one (NP) surveying data for completeness or inaccuracies.

### 2.3. Inclusion and Exclusion Criteria

Studies involving adult patients with surgically treated colorectal liver metastases and known RAS status were evaluated for inclusion in the present analysis. To be considered eligible for inclusion, at least one of the following outcomes were required to be reported: (1) OS or LS-DFS on patients undergoing anatomical versus non-anatomical resections, with outcomes reported separately for mutated-RAS (mutRAS) and wild-type RAS (wtRAS) patients, (2) OS or LS-DFS of mutRAS and wtRAS patients receiving R1 resections relative to their R0 counterparts, and (3) studies reporting on prognosticators or composite risk scores that are associated with poor OS or DFS following curative-intent liver resection.

The following set of predetermined exclusion criteria was utilized for the purposes of this study: (1) non-clinical studies, case reports, letters, reviews and editorials, (2) non-English language, (3) studies including patients that received treatments other than liver resection for primary (non-recurrent) CRLM, (4) studies reporting outcomes other than 5-year OS and/or 5-year LS-DFS, (5) studies with overlapping population datasets (in this case only the latest published study was included), (6) studies including patients with unknown RAS status, and (7) studies reporting outcomes not stratified by RAS status.

## 3. Results

A total of 669 unique abstracts were generated by the previously described systematic search. After excluding duplicate articles and those of obvious irrelevance, 58 studies were evaluated in full text for inclusion. Following the application of the exclusion criteria, a total of 10 studies were deemed eligible for inclusion in the final qualitative synthesis (Figure 1). Overall, four studies originated from North America, two from Europe, and four from East Asia. The total number of patients evaluated was 3712, of which 1427 (38.4%) were mutRAS and 1910 (60.5%) exhibited synchronous metastatic disease. The presence of extrahepatic disease spread at presentation was inconsistently reported amongst included studies (Table 1). The study by Choi et al. [18] explicitly stated exclusion of all patients with extrahepatic spread as a selection criterion. In the seven studies not reporting on the presence of extrahepatic disease, a curative-intent treatment strategy was pursued, implying absence of extrahepatic sites, but not directly reporting it. 

### 3.1. Anatomical Versus Non-Anatomical Resections

A total of four studies [16,18,19,20], incorporating 1089 patients of which 488 (44.8) were of mutRAS status, compared anatomic versus non-anatomic liver resections depending on RAS status (Table 2). Amongst them, the LS-DFS outcome was the most consistently reported metric of oncologic survival, with two studies reporting statistically significant improvement following anatomic resections in both mutRAS and wtRAS patients [16,19]. A trend towards improved outcomes with anatomic resections in wtRAS, but not in mutRAS, patients was observable in the retrospective propensity-matched (PSM) study by Joechle et al. [20]. Contrariwise, the study by Choi et al. [18] indicated worse LS-DFS outcomes in patients undergoing anatomic resections. It should be noted that the same study reported only on synchronously presenting CRLM, thus presenting an outlier study.

### 3.2. R1 Resection Rates in Mutated and Wild-Type RAS Patients

Six studies [3,14,15,18,21,22] reported the rate of R1 resection stratified by RAS status. Amongst a total of 2259 patients (38.2% of which were mutRAS), the overall R1 resection rates were 21.2% and 15.6% in mutRAS and wtRAS patients, respectively. A clear trend towards increased R1 rates in mutRAS patients was demonstrable across all studies, with R1 resection rates in the mutRAS group ranging from 3.2% to 41%, and in the wtRAS group from 2.6% to 29.6% (Table 3). In three studies [14,15,22], the impact of RAS mutations on the completeness of surgical resection was statistically significant.

### 3.3. Impact of RAS Status on Survival after Incomplete Resection

In three studies, survival of R1 resected patients was evaluated according to RAS mutational status [21,22,23]. The study by Xu et al. [22] was the only one registering statistically significant differences, with 5-year OS being 4% in mutRAS patients and 27.8% in wtRAS (Table 4). Moreover, mutRAS status conferred a 77% increase in the Hazard of death amongst patients who received a R1 resection (*p* = 0.02). In terms of 5-year overall survival, two studies reported considerably worse outcomes in mutRAS patients [22,23] while the third study by Hatta et al. [21] reported comparable survival rates. Finally, LS-DFS was reported in two studies with concurring results. In the study by Procopio et al. [23], the 5-year LS-DFS rate was four times lower in mutRAS patients, while in the study by Hatta et al. [21], multivariate analysis revealed a 21% increase in the hazard for developing a liver recurrence, albeit without statistical significance.

## 4. Discussion

The debate on the optimal margin length after resection of CRLM has been long-standing with Ekberg et al., in 1985, proposing one centimeter as the standard marginal width for CRLM resection [24]. In 2005, Pawlik et al. observed no difference in survival between patients with 1–4 mm, 5–9 mm, or more than 10 mm marginal width, suggesting that a subcentimetric margin greater than 1 mm may be an adequate histopathological goal to ensure disease clearance, especially when paired together with neoadjuvant systemic therapy [25]. Various studies published thereafter have revisited the topic and as yet, remain inconclusive on whether 1 mm or 1 cm should be the cut-off for determining an adequate resection [9,26]. Since the presence of tumor elements in a radius of 2–4 mm around the tumor has been previously reported in a small minority of patients [27,28], it becomes obvious that a one-fits-all approach does not exist but in fact tumor biological aggressiveness should dictate treatment aggressiveness as indicated by resection margin width. 

To this end, RAS mutations are useful in identifying the more invasive and aggressive cancer phenotypes, manifesting with microvascular invasion, and poor response to chemotherapy regimens [29]. Previous studies have shown resection margin as an independent predictor of poor prognosis, likely due to the presence of microscopic tumor extension away from the epicenter of the involved CRLM lesions [28,30]. In patients with mutRAS status, micrometastases have been reported to exist up to a 4 mm away from the primary nodule, and up to 1 mm in wtRAS patients [27,28,31]. Such findings are indicators that CRLM surgery should incorporate RAS as a biomarker in its decision-making process. In this context, we evaluated existing literature to better delineate the effect of RAS status on margin positivity and what implications it carries for patient survival.

We first sought to evaluate whether anatomical resections confer additional survival benefits to mutRAS patients. DeMatteo et al. observed a benefit from more aggressive approaches and extensive resections in a retrospective review published in 2000, but this study predated the use of modern perioperative chemotherapy and thus is not likely applicable in the current era [32]. In fact, neoadjuvant chemotherapy in cases of resectable CRLM has been found to confer benefits in relation to patients’ RFS and OS. Moreover, it may prove useful for prognostication purposes since tumor sensitivity to chemotherapy, as indicated by tumor downsizing, is another determinant of tumor biology that has often been associated to enhanced postoperative outcomes [33,34].

While no prospective randomized trials comparing anatomical to non-anatomical liver resections have been conducted to date, accumulating evidence indicates that non-anatomical resections may be equally effective in managing CRLM [35]. Concurrently, multiple studies have associated increases in morbidity and mortality rates with anatomical resections, tipping the balance of favor towards non-anatomical resections, which remain the standard of care to date [36]. The reluctance of performing extended anatomical resections also stems from the fact that liver parenchyma should be spared in a multirecurring disease setting, so as to permit future reinterventions as necessary. Nevertheless, the equivalence between anatomic and non-anatomic resections for CRLM has only been demonstrated in retrospective studies and more importantly, it has not been evaluated within the context of RAS mutational status. Following the findings of the present systematic review, literature on anatomical versus non-anatomical liver resections stratified in mutRAS appears heterogeneous. Margonis et al. initially reported enhanced local recurrence outcomes with anatomical resections in mutRAS status, an observation that was, nonetheless, not reproduced in subsequent studies [16,18,19,20]. Of note, two of the included studies reported on resection margins with the two techniques, indicating a trend towards a non-significant increase of R1 resection rates in mutRAS patients undergoing non-anatomic resections. Whether or not anatomical resections yield benefits in the management of patients with CRLM remains to be seen in the results of the ongoing ARMANI clinical trial (NCT04678583).

Irrespective of the extent of parenchymal resection, R0 resection patients tend to fare better in the long-term [37]; however, it is uncertain whether that benefit extends to all patients regardless RAS status. Nishioka et al. recently reported that tumor biology rather than R1 resection is the main determinant of local recurrence and long-term survival rate [38]. This view is rational, considering the advent of perioperative chemotherapy that serves to further sterilize the postresection surgical field. In this regard, expanding the application of liver resections at the expense of increased postoperative R1 rates becomes an interesting prospect. In the present analysis, we pooled available evidence reporting the survival of patients with R1 status according to the presence of RAS mutations. Literature was sparse and inconsistent, with two studies reporting four times lower 5-year survival rates in mutRAS patients and in another one, equivalent results between the compared groups (Table 3). While an R1 resection in RAS mutated CRLM is associated with a poor prognosis, it is not clear whether survival can be substantially improved by employing a more aggressive surgical strategy. Previous studies suggested that R1 resection margins may not matter at all for the mutRAS subpopulation [39]; however, whether this finding is reproducible or generalizable remains, as yet, debatable. 

The hypothesis that RAS mutations are linked to a more aggressive disease subtype that resists eradication is further supported by the observed R1 resection rates in mutRAS patients, which are considerably higher relative to the ones registered in their wtRAS counterparts. Specifically, pooled R1 resection rates amongst mutRAS patients were found to be 21.2% versus 15.6% in wtRAS patients. Moreover, three of the six evaluated studies reported a statistically significant difference clearly in favor of wtRAS status (Table 3). This is a very interesting observation denoting that radical resection is less likely in mutRAS patients when conventional curative-intent resectional treatments are pursued. Zhang et al. postulated that micrometastatic spread away from the primary lesion is the underlying mechanism explaining the discrepancy between observed R1 resection rates between mutated and wild-type RAS patients [15]. Indeed, in their analysis, the authors found that micrometastases more frequently accompanied mutRAS status and additionally, they exhibited significantly wider spatial spread. Interestingly, preoperative chemotherapy significantly reduced the average maximum distance of micrometastatic spread regardless of RAS status. Such observations carry important implications with respect to surgical strategy. Extending resection margins in mutRAS patients thus represents a rationale choice. Whether this may be accomplished by performing a formal anatomical resection or simply by increasing the width of parenchymal resection in non-anatomical hepatectomies is not yet clear. Evidently, tumor biology, especially in terms of RAS status, is an important determinant of the applicability of resectional treatments, and should complement existing postoperative survival prognosticators, such as R1 status, to guide further treatment perspectives and optimize patient outcomes.

The changing landscape of CRLM biology mandates that composite decision-making tools be applied to clinical practice in order to identify patients who derive benefit from surgical resection. Passot et al. in 2017 proposed lymphatic spread of the primary tumor, size greater than 3 cm, and history of at least 7 cycles of chemotherapy as important indicators of poor survival after liver resection in mutRAS patients [40]. The observed 5-year OS was 0% for patients with all three risk factors, prompting the investigators to suggest avoidance of aggressive surgical treatment in favor of systemic chemotherapy. A similar subsequent attempt by Brudvik et al. utilizing similar predictive factors observed equivalent outcomes in high-risk patients undergoing surgery [41]. The common denominator between these studies is the presence or absence of RAS mutations, with BRAF presenting an interesting alternative biomarker, albeit one with considerably smaller mutational frequency in CRLM [42]. It therefore becomes increasingly clear that better patient stratification and optimization of treatment allocation is necessary in the current era of precision surgery for metastatic colorectal cancer.

As evidence continues to accumulate, attention has also been brought towards the diversity of RAS mutations and the prognostic implications they carry. Interestingly, the majority of detected mutations in patients with CRC involve codons 12 and 13, with the most commonly encountered point mutations being 12 glycine to valine (G12V), 12 glycine to aspartate (G12D), 12 glycine to serine (G12S), and 13 glycine to aspartate (G13D) [43]. Early reports hinted that different mutations are associated with more aggressive disease subtypes, with codon 12 mutations in particular being linked to adverse oncological outcomes [44,45]. This insight adds another layer of complexity to the existing knowledge on the impact of RAS mutational status in CRLM. Indeed, follow-up cohort studies suggested that codon 12, but not codon 13, mutations were associated with a detriment to OS after curative intent hepatectomy [46]. Interestingly, the investigators observed that G12V and G12S were the point mutations that signified the worse OS outcome postoperatively. On the other hand, mutations in codon 13 have been associated with increased post-interventional recurrence rates, following liver resection and radiofrequency ablation [47]. While the clinical significance of specific point mutations has not yet been fully elucidated, it serves as another indicator that tumor biology is a major determinant of surgical efficacy in controlling cancer spread and consequently patients’ survival.

Although the data presented herein support the adoption of a more individualized surgical approach depending on prognostic biomarker status, several shortcomings are appreciable in existing literature. First and foremost, conflicting results for various studies more often than not stem from imbalances in patient and tumor baseline characteristics. While this may be partially offset by adopting a randomized or propensity score-matched design in future studies, it should be noted that patients with CRLM are a very heterogeneous population. In terms of RAS status, accumulating evidence implies that mutRAS and wtRAS should be evaluated separately as they exhibit considerably different disease courses. In addition, aside from the sparsely available literature on the topic, outcome reporting greatly varied from study to study, thus making direct comparisons difficult. LS-DFS arguably represents the most accurate direct metric of success of surgical treatment, as its main aim is to control the local disease component. Such outcomes were inconsistently reported amongst included studies and thus uniformity in outcome reporting should be encouraged in the future.

## 5. Conclusions

In conclusion, surgical resection is the most potent curative-intent treatment available in the armamentarium of physicians treating CRLM. However, metastases from CRC are very heterogeneous in terms of behavior, with biology largely dictating disease aggressiveness. RAS mutations are frequently found in CRLM and have been associated with suboptimal disease outcomes. After systematically reviewing existing literature on the impact of RAS status on survival after resection for CRLM, we identified a possible hidden association between R1 resection rates and RAS mutations. This may be due to the increased propensity of mutRAS tumors to exhibit micrometastatic spread that compromises resectional margins. In turn, the presence of RAS mutations may be an indicator of the need for more aggressive resectional treatments. To this end, anatomical resections may be employed, in an effort to ensure negative margins and radical disease clearance. Existing evidence is, to date, inconclusive on the role of anatomical resections in mutRAS patients; however, the need for wider resection margins is beginning to become more obvious. Moreover, certain subgroups of patients with mutRAS status derive no oncological benefit from surgery, while in others, R1 status bears no impact on their postoperative long-term survival. Although many questions remain, as yet, unanswered, the role of RAS as putative biomarker has consistently proven to be pivotal in devising patient stratification schemes, which allow surgical treatment allocation in a precise and individualized fashion.

## Figures and Tables

**Figure 1 cancers-15-02083-f001:**
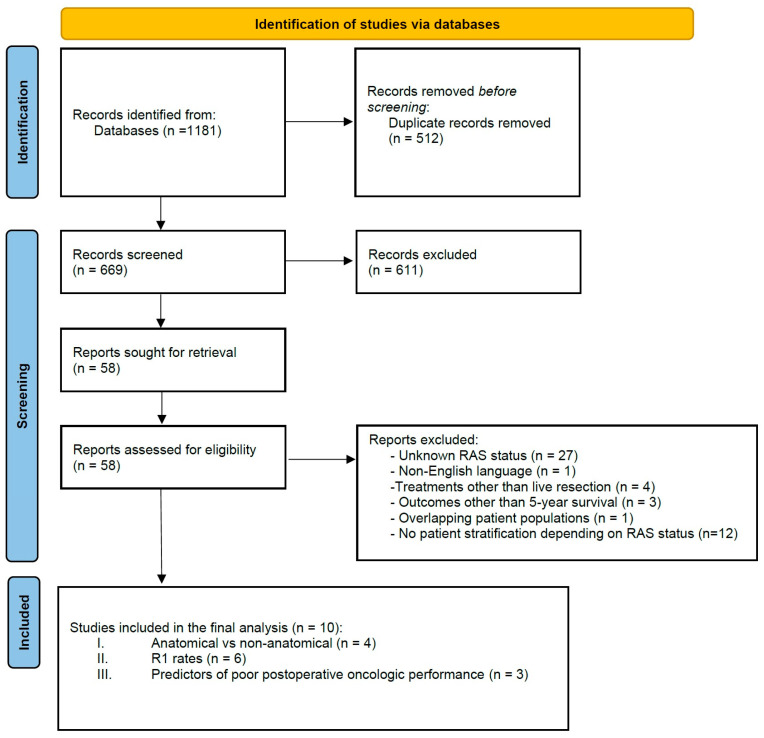
Prisma flowchart of study selection process.

**Table 1 cancers-15-02083-t001:** Baseline study characteristics and patient demographics.

Study	Year of Publication	Country of Origin	Type of Study	Total Number of Patients	mutRAS, n (%)	Synchronous Presentation, n (%)	Number of Tumors (Median, Range)	Tumor Size (Median, Range)	Extrahepatic Disease, n (%)
Choi et al. [18]	2022	Korean	Retrospective	250	94 (37.6)	250 (100)	2.85 ± 3.00 *	2.37 ± 1.80 *	0
Kawai et al. [19]	2022	Japan	Retrospective	290	104 (35.9)	167 (57.6)	1.6 ± 1.7 *	2.5 ± 2.5 *	NR
Joechle et al. [20]	2019	USA	Retrospective, PSM	360	150 (41.7)	253 (70)	1 (1–9)	2 (0.1–9.5)	55 (15)
Margonis et al. [16]	2017	USA	Retrospective	389	140 (35.9)	223 (57.3)	2 (1–3)	2.5 (1.6–4.0)	NR
Brudvik et al. [14]	2016	USA	Retrospective	633	229 (36.2)	446 (70.5)	NR	NR	NR
Zhang et al. [15]	2020	China	Retrospective	251	130 (51.8)	61 (24)	NR	NR	28 (11.2)
Hatta et al. [21]	2020	UK	Retrospective	500	152 (30.4)	233 (51.7)	2 (1–3)	3 (2–5)	NR
Margonis et al. [3]	2016	USA	Retrospective	485	178 (36.7)	277 (57.1)	2 (1–3)	2.5 (1.5–4)	NR
Xu et al. [22]	2019	China	Retrospective	214	100 (46.7)	NR	NR	NR	NR
Procopio et al. [23]	2020	Italy	Retrospective	340	150 (44.1)	NR	NR	NR	NR

* Values are presented as mean and standard deviation; NR = not reported.

**Table 2 cancers-15-02083-t002:** Survival outcomes in anatomical versus non-anatomical liver resections depending on RAS mutational status.

*Anatomic vs. Non-Anatomic*						
	*mutRAS*			*wtRAS*		
Study	LS-DFS Hazard Ratio	5-Year LS-DFS (%)	*p*-Value	LS-DFS Hazard Ratio	5-Year LS-DFS (%)	*p*-Value
Choi et al. [18]	1.23 (0.64–2.39)	NR	0.52	1.41 (0.86–2.32)	NR	0.17
**Kawai et al. [19]**	**NR**	**NR**	**0.23**	**0.42 (0.25–0.72)**	**NR**	**0.001**
Joechle et al. [20]	NR	16 vs. 17.3	0.4	NR	22.9 vs. 14.3	0.88
**Margonis et al. [16]**	**0.37 (0.21–0.64)**	**13 vs. 0**	**<0.001**	**0.63 (0.42–0.94)**	**15.9 vs. 4.3**	**0.02**

NR = not reported.

**Table 3 cancers-15-02083-t003:** Margin positivity (R1) depending on RAS mutational status.

	*mutRAS*		*wtRAS*		
Study	Total Patients	R1, *n* (%)	Total Patients	R1, *n* (%)	*p*-Value
**Brudvik et al.** [14]	**229**	**26 (11.4)**	**404**	**22 (5.4)**	**0.007**
Choi et al. [18]	94	3 (3.2)	156	4 (2.6)	0.55
Hatta et al. [21]	146	42 (28.8)	284	84 (29.6)	0.88
Margonis et al. [16]	178	35 (19.7)	307	70 (22.8)	0.49
**Xu et al.** [22]	**100**	**41 (41)**	**114**	**26 (22.8)**	**0.005**
**Zhang et al.** [15]	**117**	**26 (21.5)**	**130**	**12 (9.2)**	**0.007**

**Table 4 cancers-15-02083-t004:** Impact of RAS mutational status on survival after R1 resection.

	*mutRAS*	*wtRAS*		*mutRAS*	*wtRAS*			
Study	5-Year OS (%)		*p*-Value	5-Year LS-DFS (%)		*p*-Value	OS Hazard Ratio	LS-DFS Hazard Ratio
Procopio et al. [23]	6.8	26.9	NR	1.7	4.8	NR	NR	NR
**Xu et al.** [22]	**4**	**27.8**	**0.02**	**NR**	**NR**	**0.12**	**1.77 (1.08–2.88)**	**NR**
Hatta et al. [21]	55.5	56.3	0.57	26.8	35.8	0.15	1.08 (0.77–1.52)	1.21 (0.93–1.58)

NR = not reported.

## Data Availability

Data are available upon reasonable request.

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
