# Peer review of "Precision Surgery of Colorectal Liver Metastases in the Current Era: A Systematic Review"

_cancers, 2023, doi:10.3390/cancers15072083_

Round 1

Reviewer 1 Report

Interesting article on a hot topic. The authors have conducted an easy to read review according to the PRISMA guidelines. The proper patient selection is of paramount importance in order to achieve optimal results in terms of survival after hepatic resection for colorectal cancer liver metastases. The authors have assessed primarily the RAS mutational status as prognostic factor. However, the impact of other parameters on survival as well such as the response to neoadjuvant chemotherapy should be included in the discussion section of the manuscript. In addition, the diversity of RAS mutations and its prognostic implications should be further discussed. Reference list should be updated and expanded with the most recent literature data.

Author Response

We thank the reviewer for his/her time taken reading our manuscript. We are also grateful for the feedback, interesting comments and the opportunity to respond.

Our purpose for the present systematic review was to focus on the paradigm shift that the global surgical community has witnessed in recent years; colorectal liver metastatic disease was converted from a purely surgical disease with abysmal prognosis to one managed by a multidisciplinary team, with a patient-tailored approach. The term “precision surgery” in the title serves as a testament to this paradigm shift, both in terms of enhanced surgical technical accuracy by the introduction of newer technologies but also in terms of more accurate patient selection which allows optimal utilization of surgical treatments, while minimizing unwarranted morbidity.

In the present review, we elected RAS as the main point of focus to guide precision surgery for CRLM, however, as the reviewer very accurately points out there are other significant determinants of the biologic behavior of metastatic colorectal tumors. We chose RAS as a marker to guide precision surgery because more literature was available for evaluation, relative to other factors (such as the ones indicated by the reviewer, i.e. response to chemotherapy and specific RAS point mutations). We agree with the reviewer that despite the scarcity of data on these factors, the discussion should be expanded to include them as well for the sake of scientific completeness. We have thus amended the text of the discussion as follows to include a comment on neoadjuvant chemotherapy response as a determinant of tumor biology: “In fact, neodjuvant chemotherapy… enhanced postoperative outcomes”. Additionally, a discussion on the diversity of RAS point mutations and their clinical significance, along with relevant literature was added (“As evidence continues to accumulate…patients’ survival”).

Reviewer 2 Report

the paper is good showing the importance of free margin resection and introduce the quality of type of resection R1 and compare with genetic biomarkers

Author Response

We thank the reviewer for his/her time taken to go through our manuscript. We are also grateful for the kind comments.